# Water-Based and Land-Based Exercise for Children with Post-COVID-19 Condition (postCOVIDkids)—Protocol for a Randomized Controlled Trial

**DOI:** 10.3390/ijerph192114476

**Published:** 2022-11-04

**Authors:** Anna Ogonowska-Slodownik, Marta Kinga Labecka, Katarzyna Kaczmarczyk, Renae J. McNamara, Michał Starczewski, Jan Gajewski, Agnieszka Maciejewska-Skrendo, Natalia Morgulec-Adamowicz

**Affiliations:** 1Faculty of Rehabilitation, Jozef Pilsudski University of Physical Education in Warsaw, Marymoncka 34, 00-968 Warsaw, Poland; 2Physiotherapy, Prince of Wales Hospital, Sydney, NSW 2031, Australia; 3Faculty of Medicine and Health, Sydney School of Health Sciences, The University of Sydney, Sydney, NSW 2006, Australia; 4Woolcock Institute of Medical Research, Sydney, NSW 2037, Australia; 5Faculty of Physical Culture, Gdansk University of Physical Education and Sport, 80-336 Gdansk, Poland; 6Institute of Physical Culture Sciences, University of Szczecin, 70-453 Szczecin, Poland

**Keywords:** fatigue, long COVID, SARS-CoV-2, quality of life

## Abstract

The most common symptoms of post-COVID-19 condition in children are fatigue, shortness of breath, exercise intolerance, and weakness. The post-COVID-19 condition in children can be very debilitating and lead to prolonged school absences, high morbidity, and limitations in daily functioning. The aim of this research project is to determine the effectiveness of land-based and water-based exercise interventions on exercise capacity, fatigue, health-related quality of life, and pulmonary function in children with post-COVID-19 condition. This study is a prospective randomized controlled study with pre- and post-intervention assessment. Participants will be recruited from Warsaw’s primary schools and primary healthcare units according to the inclusion criteria: (i) symptoms of post-COVID-19 condition lasting more than one month following initial COVID-19 infection confirmed by the diagnosis by general practitioner (including obligatory fatigue and shortness of breath/respiratory problems); (ii) age 10–12 years old. Participants meeting the inclusion criteria will be randomized to one of three groups: water-based exercise, land-based exercise, or control (no exercise). We hope this study will provide guidance for long-COVID-19 rehabilitation in children.

## 1. Introduction

During the early stages of the COVID-19 pandemic, fatality was a major concern and area of intense investigation and intervention; however, this has now progressed to recognition of the physical and psychosocial post-acute consequences of COVID-19, called post-COVID-19 condition [1]. Post-COVID-19 condition is defined as occurring at least four weeks after the onset of COVID-19, with symptoms that last for at least two months and cannot be explained by an alternative diagnosis. Post-COVID-19 condition is a frequently occurring condition in children [2,3]. The most common symptoms of post-COVID-19 condition in children are fatigue, shortness of breath, exercise intolerance, weakness, muscle and joint pain, headache, insomnia, respiratory problems, and palpitations [4]. In a recent study of children post-COVID-19 infection, almost half presented with at least one symptom of post-COVID-19 condition over 60 days after initial infection [4]. Older age, muscle pain on hospital admission, and intensive care unit admission are significantly associated with experiencing post-COVID-19 condition in children [2]. The post-COVID-19 condition in children can be very debilitating and lead to prolonged school absences, high morbidity, and limitations in daily functioning [5,6]. Although children generally present with mild, acute COVID-19, they are at risk of prolonged organ damage, similar to what has been identified in adults [7]. It is important that long-term sequelae in children are not dismissed as psychological problems and to provide them with comprehensive diagnostic assessments, care, and support [8]. The basic biological mechanisms responsible for these debilitating symptoms are still unclear [9]. The COVID-19 pandemic profoundly affected young children’s development through loss of time in education, isolation, an increase in poverty and food insecurity, loss of caregivers, and heightened stress [2]. Further, there is also increasing evidence that restrictive measures aimed at limiting the pandemic had a significant impact on children’s mental health [4].

Rehabilitation after COVID-19 has the goal of improving respiratory symptoms, preserving function, and reducing complications and disability, and has a positive effect on the psychological sphere, reducing anxiety and depression [10]. A recent review highlighted the need to consider community-based pulmonary rehabilitation programs for COVID-19 survivors with compromised lung function [11]. People experiencing post-COVID-19 condition need support to manage their symptoms, especially fatigue, while also helping them safely pursue the potential benefits of exercise [12]. Organizational and service delivery issues have also arisen during the COVID-19 epidemic, which has altered face-to-face visits, with consultations being mostly in the form of online consultations [13]. Virtual group-based consultations may be a challenge for children. Interaction and non-verbal communication in the relationship between a young patient and their health professional are very important [14].

The range, severity, frequency, and duration of symptoms associated with post-COVID-19 condition in children present significant challenges for the precise recommendations for rehabilitation exercise training [12]. Guidelines provided by the American Academy of Pediatrics to assist in the management of return to sport and physical activity for children recovering from COVID-19 illness recommend that children who have asymptomatic disease or a mild form of the disease need to be screened by their primary care providers before returning to play. All children who have had COVID-19 should be asked about chest pain, shortness of breath, palpitations, or fainting [15]. A minimum of ten days of rest after the disease, including seven days without symptoms, and no cardiorespiratory symptoms when performing normal activities of daily living are advised [16].

The aquatic environment offers an opportunity to support and positively influence exercise in children. In children with respiratory conditions, the hydrostatic pressure provided by water improves the efficiency of the cardiorespiratory system [17]. Aquatic exercises also improve pulmonary function in children with obesity [18] and reduce the level of fatigue and improve the quality of life in children with juvenile dermatomyositis [19]. Specific breathing programs in water can improve lung function in adolescents with scoliosis [20]. In relation to the safety of the aquatic environment in the COVID-19 pandemic, research has demonstrated that chlorinated water reduces the SARS-CoV-2 infectious titer by at least three orders of magnitude [21]. Land-based exercise interventions are also effective in children with respiratory conditions. Children with asthma perceived that their fitness and asthma had improved and reported an increased health-related quality of life after six weeks of active play exercise intervention [22]. Eight weeks of regular submaximal exercise had beneficial effects on the quality of life and exercise capacity in children with asthma [23].

COVID-19 can have a long-term impact in children, and there is a need to implement measures to reduce the impact of post-COVID-19 condition on a child’s health and to ensure children receive appropriate therapeutic interventions. In particular, greater clarity and tailoring of exercise-related advice for children with post-COVID-19 condition and improved support to resume activities important to individual well-being is warranted. To the best of our knowledge, there are no studies describing exercise programs for children with post-COVID-19 condition.

### Objectives

The primary aim of this research project is to determine the effectiveness of land-based and water-based exercise interventions on exercise capacity and fatigue in children with post-COVID-19 condition.

The secondary aims are to determine the effectiveness of land-based and water-based exercise on health-related quality of life and pulmonary function in children with post-COVID-19 condition, and to compare outcomes to norms for healthy children.

## 2. Materials and Methods

This study is a prospective randomized controlled study with pre- and post-intervention assessment. The protocol was developed in accordance with the guidelines and checklists for Standard Protocol Items: Recommendations for Interventional Trials (SPIRIT) [24]. Results will be reported as stated in the Consolidated Standards of Reporting Trials (CONSORT) statement [25]. The study is being conducted at the Faculty of Rehabilitation, Jozef Pilsudski University of Physical Education in Warsaw (AWF Warsaw), Poland. The study has been registered on clinicaltrials.org (NCT05216549; 31 January 2022) and is in compliance with the latest version of the Helsinki Declaration [26]. The study was approved by the Ethics Committee of the AWF Warsaw. Written consent for participation will be expressed by the legal guardians of the children.

### 2.1. Study Population

Participants will be recruited from Warsaw’s primary schools and primary health-care units according to the inclusion criteria: (i) symptoms of post-COVID-19 condition lasting more than one month following initial COVID-19 infection confirmed by the diagnosis by general practitioner (including obligatory fatigue and shortness of breath/respiratory problems); (ii) age 10–12 years old. As it has been suggested that a positive COVID-19 test is not required for a diagnosis of post-COVID-19 condition [27], there will be an optional inclusion criteria: positive RT-PCR test and/or positive result in test for antibodies against the SARS-CoV-2 coronavirus 1–8 months prior to the study start. The exclusion criteria will consist of: absolute contraindications to exercise; unstable cardiac conditions; currently engaged in regular exercise training more than twice per week. The necessary minimum total number of participants (n = 84) was obtained using the G*Power program assuming detection of a moderate effect size of REPETITIONxGROUP interaction (d = 0.50 or eta square 0.06) for exercise capacity and fatigue with a significance level of 0.05 and statistical power of 0.85.

### 2.2. Randomization

Participants meeting the inclusion criteria will be randomized to one of 3 groups: water-based exercise, land-based exercise, or control (no exercise). The randomization sequence will be designed by an investigator external to the study using an Excel random number generator. Group allocation will be prepared in 5 random versions. Finally, the allocation providing the lowest F value for MANOVA comparing basic characteristics of the study subjects will be used. Randomization will be stratified according to age, sex, and the level of exercise capacity. Concealed allocation will be achieved using opaque envelopes. Due to the nature of exercise interventions, it will not be possible to blind the physiotherapist or participants. 

### 2.3. Outcome Measures

Outcome measures will be collected at baseline and immediately after the 8-week intervention. Table 1 shows the distribution of the different measures across the time points during the study (SPIRIT table).

#### 2.3.1. Demographic Data

Demographic data that will be collected include: age, sex, date of COVID-19 onset, current physical activity level (participation in organized physical activity per week), swimming skills, and symptoms of post-COVID-19 condition. Height and weight will be measured before and after the intervention.

#### 2.3.2. Diagnosis of General Practitioner

Participants will be required to undertake a pre-participation screening for exercise testing and training with the general practitioner including: a medical history questionnaire, general health screen, blood pressure, medication use, nutritional assessment, heat- and hydration-related risk factors, mental health considerations, tests for SARS-CoV-2 antibody only for participants without positive RT-PCR test and/or positive result in test for antibodies against the SARS-CoV-2 coronavirus in the 1–8 months prior to the study start.

#### 2.3.3. Exercise Capacity

Exercise capacity will be measured using the modified Balke treadmill protocol previously validated on a large population-based sample of children aged 7–18 years [28]. After 30 min of the seated rest, to familiarize participants with the treadmill speed, the first three minutes of the entire protocol will consist of walking on the flat without an incline with the speed set at 5.3 km/h. After the first three minutes, the incline of the treadmill will increase to 6%, and afterward by 2% each minute until 22%. After reaching 22%, the angle of the treadmill will be constant and, at each subsequent minute, the treadmill speed will be increased by 0.5 km/h until volitional exhaustion. 

The validated Pictorial Children’s Effort Rating Table (PCERT) will be used to measure exhaustion during exercise capacity assessments [29]. The PCERT is a visual assessment scale suitable for children to effectively self-report the perceived rate of exertion during physical activity by means of a verbal, numerical, and pictorial representation scale. The scoring is based on a scale from 1 to 10, where 1 is “very, very easy” and 10 is “so hard I’m going to stop” [30]. Participants will be asked to rate their perceived exertion during the treadmill test with use of the PCERT: before the test, at the 5th minute of the test, and immediately at the end of test. Standardized verbal encouragement will be given throughout the test. Heart rate (HR) will be monitored continuously during the test using the POLARV800 (Polar Electro OY Finland). Breathing parameters will be measured by the “breath-to-breath” method using the Cortex MetaMax 3B ergospirometer system (Biophysik GmbH, Leipzig, Germany). The gas analyses will be sampled at 15 s intervals. It is expected that most participants will not achieve maximal oxygen uptake (VO_2_max) values as described by a plateau in uptake of oxygen, respiratory exchange ratio (RER) > 1.5, HR 220—age or rating in PCERT scale 9 or 10. The peak oxygen uptake values will be established from the highest-achieved-intensity 15 s samples. The treadmill test will be terminated for safety using the following criteria: HR > 200 bpm, RER > 1.1, dyspnea or neurological symptoms.

It is planned to calculate Oxygen-Uptake Efficiency Slope (OUES) from the collected data. The OUES utilizes data from the entire duration of a graded exercise test; however, it is not necessarily dependent on achieving a maximal exercise response as is required when VO_2_max is assessed [31,32]. OUES is unbiased by maximal effort, test duration, or engagement of the patient, which can be challenging in a population of children. The OUES represents how effectively the oxygen is extracted by the lungs and used in the periphery and is derived from the logarithmic relation between oxygen uptake and minute ventilation [33]. OUES values, absolute or indexed to weight, height, age, body surface area, or percentage of predicted, can be used. Body surface area will be calculated using the equation of Haycock et al. [34], validated for children.

#### 2.3.4. Fatigue

Fatigue will be measured using the Polish adaptation of the Cumulative Fatigue Symptoms Questionnaire (CFSQ) validated by Kulik and Szewczyk [35]. This questionnaire measures the level of chronic fatigue and severity of individual symptoms including general fatigue, decreased vitality, mental overload, somatic symptoms, anxiety, and discouragement. It includes 30 statements with responses of “never”, “sometimes”, and “often”, where the responses are assigned the values of 0, 1, and 2, respectively. The response scores are added together and the CFSQ total score ranges from 0 to 60; the higher the score, the greater the severity of the symptoms of chronic fatigue [35]. Scores are categorized as low (0–14), average (15–30), high (31–45), and very high (46–60). The more detailed characteristics of fatigue will be explored by adding up the scores in the subscales: general fatigue, weakened vitality, psychological overload, somatic symptoms, anxiety, and discouragement about studying and school.

#### 2.3.5. Health-Related Quality of Life

Health-related quality of life (HRQOL) will be measured with the Polish version of the Pediatric Quality of Life Inventory 4.0 Generic Core Scales (PedsQL^TM^ 4.0) for children and for parents. The Polish version of PedsQL^TM^ 4.0 was validated by Talarska et al. [36]. PedsQL^TM^ 4.0 is a brief, standardized, generic assessment instrument that systematically assesses perception of HRQOL in healthy children and those with acute and chronic health conditions. It is based on a modular approach to measuring HRQOL. The questionnaire contains 23 questions divided into the following subsections: physical functioning (8 items), emotional functioning (5 items), social functioning (5 items), and school functioning (5 items). The sum of these areas is the overall quality of life (total score). The responses are reported on a 5-point Likert scale ranging from 0 (no problem) to 4 (almost always), with a total score ranging from 0 (best HRQOL) to 100 (worst HRQOL) [37]. Further, two summarized functional children’s areas will be reported: the Physical Health Summary Score covering the area of Physical Functioning, and the Psychosocial Health Summary Score will be taken into account for analysis. Both participants and their parents will complete the questionnaire separately. It is planned to compare children’s and parents’ perception of the HRQOL of the child with post-COVID-19 condition. 

#### 2.3.6. Pulmonary Function

Measurements will be taken in accordance with the guidelines of the European Respiratory Society and American Thoracic Society [38] and regarding the indications for the execution of respiratory function tests in the COVID-19 era [39]. During pulmonary function assessment, we will record the following parameters: VC (vital capacity), FVC (forced vital capacity), FEV_1_ (forced expiratory volume in 1 s), MEF_75_ (maximal expiratory flow at 75% of forced vital capacity), MEF_50_ (maximal expiratory flow at 50% of forced vital capacity), and MEF_25_ (maximal expiratory flow at 25% of forced vital capacity) using the portable Lungtest Handy (MES LLC, Cracow, Poland).

Further, the results of pulmonary parameters in the form of absolute values for FVC, VC, FEV_1_, MEF_25_, and data on race, sex, age, and height of the participant will be entered into the GLI 2012 (Global Lung Function Initiative, Berlin, Germany) calculator [40], yielding a percentage of the norm for healthy children. On this basis, the following parameters will be calculated: FEV_1_pred, FEV_1_%pred, FEV_1_%tile FEV_1_LLN, FEV_1_Z, VCpred, VC%pred, VC-LLN, VC Z, FEV_1_/VC, FEV_1_/VCpred, FEV_1_/VC%pred, FEV_1_/VC%tile, FEV_1_/VC-LLN, FEV_1_/VC Z, FVCpred, FVC%pred, FVC-LLN, FVC Z, FEV_1_/FVC, FEV_1_/FVCpred, FEV_1_/FVC%pred, FEV_1_/FVC%tile, FEV_1_/FVC-LLN, FEV_1_/FVC Z, MEF_25_pred, MEF_25_%pred, MEF_25_%tile, MEF_25_LLN, and MEF_25_ Z (where “pred” stands for predicted value, “%pred” for percentage of predicted value, “%tile” for percentile, LLN for lower limit of normal, and “Z” for Z score; the number of standard deviations the measured value differs from the predicted value).

### 2.4. Intervention Protocol

The water-based and land-based exercise training will be conducted twice a week, 45 min per session, for eight weeks. The exercises will be supervised by an experienced physiotherapist. 

Exercises in the water and on land will be matched in terms of intensity, duration, and muscle groups trained. Each session will consist of warm-up (8 min), aerobic training (32 min), and cool down (5 min). During the warm-up, participants will perform upper and lower limbs aerobics, breathing exercises, and stretches. The aerobic training component will consist of two circuit stations with endurance exercises for upper and lower limbs and a focus on breathing patterns. In each circuit, there will be five stations with different exercises. The activity at each station will last one minute, with 15 s rest periods in-between. At each station, the exercise will be performed by two participants together for motivation. The water-based and land-based exercises protocols are presented in Table 2. 

The aerobic training component will also include playing games (Table 3). The cool down will consist of upper-limb and thoracic cage stretches, and breathing control.

Participants will be encouraged to exercise at an intensity of 6–8 (“getting quite hard” to “hard”) on the Pictorial Children’s Effort Rating Table (PCERT) [29]. Training intensity will be measured twice during the aerobic exercise training, during each circuit. If participants report an exercise intensity below 6 on the PCERT, they will be encouraged to increase the intensity by increasing their speed and/or range of motion. After the first month of exercises, in order to make the classes more attractive and maintain exercise intensity, music and friendly competition between the participants will be introduced. Games will vary at each exercise session. Equipment used in the water-based exercise sessions will consist of floatation “noodles”, and that in the land-based exercise will consist of hand-held dumbbells, medicine balls, cones, and sashes. 

The land-based exercise sessions will be conducted in a gymnasium with a controlled temperature at the AWF Warsaw. The water-based exercise sessions will be conducted in a swimming pool (depth 1.2–1.5 m; length 12 m; width 3 m; water temperature approximately 30 °C; humidity 60%) at a fitness club in Warsaw, and participants will be encouraged to exercise at a depth where the water level sits between the clavicle and xiphisternum. 

The control (no exercise) group will be advised to not alter their current exercise routine and will be offered the option of joining a water-based or land-based exercise program after the final assessment has been completed.

### 2.5. Procedure

Participants will be recruited from Warsaw’s primary schools and primary healthcare units through various channels (flyers, e-mails, Facebook posts etc.) where information with a link and QR code to an online eligibility pre-screening questionnaire will be provided. The pre-screening will address the specific inclusion and exclusion criteria for the study, and basic demographic data. Concurrently, the eligible participants (children with parents) will be invited to undertake the main screening and diagnosis of a general practitioner at the Faculty of Rehabilitation, AWF Warsaw. The purpose of the study will be explained to the children and parents. After obtaining consent, the general practitioner will ensure all inclusion and no exclusion criteria have been met. Participants who meet the inclusion criteria will be invited to attend a baseline assessment at the Central Research Laboratory at AWF Warsaw. 

At the baseline assessment session, participants and parents will be requested to complete the PedsQL^TM^ 4.0 and CFSQ. The instructions will be explained by the principal researcher. In case of any doubt, parents and children will be able to ask the principal researcher for help. The research team will measure the participant’s weight and height and calculate body mass index (BMI). The general practitioner will perform an examination of the participants to determine any contraindications to exercise testing. Pulmonary function and exercise capacity testing will be performed. 

After completion of the baseline assessment, participants will be randomly assigned into one of three groups (water-based exercise, land-based exercise, or control), and simultaneous implementation of the two exercise programs will commence. The attendance rate will be recorded by the physiotherapists after each session. Children will also have attendance cards on which they will collect stamps to increase motivation to attend. After 8 weeks, all assessment measures for all participants will be repeated in the Central Research Laboratory at AWF Warsaw.

### 2.6. Statistical Analysis

Normality of the distributions of the quantitative variables under study will be tested using the Shapiro–Wilk test. Data will be presented as mean ± SD where appropriate. The mixed-design ANOVA will be used to compare means. The interaction of fixed and repeated factors will be analyzed. Tukey’s post hoc test will be used for detailed comparisons. For variables that do not meet the criteria of normality of distribution, data will be presented as medians and quartile range. The Mann–Whitney U test will be used to compare the groups. The variables as well as their increments will be analyzed. Effect sizes will be assessed by partial eta squared (ANOVA) or the Glass rank-biserial correlation coefficient (the Mann–Whitney U test). The agreement of children’s and parents’ responses in the PedsQLTM 4.0 questionnaire will be analyzed with the compatibility factor Kappa-k. All statistical calculations will be conducted using the Statistica 14.0.0.15 program (TIBCO Software Inc., Palo Alto, CA, USA, 2020).

### 2.7. Missing Data

Missing data may occur as the result of a discontinuation of participation in the program or due to unexpected random events (for example, illness). As missing cases are not expected to exceed 10% of the initial number of subjects, data will be analyzed as complete case analysis, excluding the missing data from the analysis.

## 3. Discussion

This study has been designed to investigate the effect of water-based and land-based exercise training on the exercise capacity, fatigue, health-related quality of life, and pulmonary function of children with post-COVID-19 condition. After the acute phase of SARS-CoV-2 infection, children can develop long-COVID symptoms. Children aged 0–14 years who had a SARS-CoV-2 infection had more prevalent long-lasting symptoms [41]. The prevalence of post-COVID-19 condition among children and adolescents was 25%, and the most prevalent clinical manifestations were mood symptoms, fatigue, and sleep disorders [42]. The most common post-COVID-19 symptoms in children/adolescents are fatigue, lack of concentration, and muscle pain [3]. Returning to physical activity should be gradual and preceded by an accurate physical examination in young subjects previously affected by the coronavirus disease [43]. Nevertheless, as physical exercise has been found to stabilize the autonomic nervous system, exercise therapy might be a safe and effective remedy for the post-COVID-19 condition [44]. The American Academy of Pediatrics released a *COVID-19 Interim Guidance: Return to Sports and Physical Activity*, updated in September 2022 with different recommendations on the return to youth sports and activity after COVID-19 depending on three categories describing the severity of COVID-19, but not addressing groups of children and adolescents with post-COVID-19 symptoms [45]. Although it is clearly stated that the return should be gradual and safe, there is no recommended guide to the intensity or type of exercises.

Research examining the impact of water- or land-based exercise training in healthy children and in children with neuromuscular disorders, obesity, and asthma has demonstrated improvements in pulmonary function [18,46,47]. However, there is no consensus regarding the most optimal mode of exercise training to increase pulmonary function and exercise capacity in children. One recommendation is a 20 min duration with a frequency of two sessions per week at a moderate intensity to increase maximal oxygen consumption [48]. To the best of our knowledge, this is the first study investigating the effect of exercise training on reducing post-COVID-19 symptoms in children. 

In adults with post-COVID-19 condition, many of the recommendations for exercise training have been extrapolated from evidence for people with chronic respiratory disease due to the limited evidence in people with post-COVID-19 condition. Evidence has demonstrated that exercise programs composed of resistance exercise (e.g., 1–2 sets of 8–10 repetitions at 30–80% of 1 RM) along with aerobic exercise (e.g., 5 to 30 min at moderate intensity) may improve the functional capacity and quality of life in post-COVID-19 adult patients [49]. In patients with post-COVID-19 condition, exercise capacity, functional status, dyspnea, fatigue, and quality of life improved after 6 weeks of personalized interdisciplinary pulmonary rehabilitation [50]. Adult individuals with post-COVID-19 condition that completed a 6 weeks, twice-a-week supervised rehabilitation program demonstrated statistically significant improvements in exercise capacity, respiratory symptoms, fatigue, and cognition [51]. The program comprised of aerobic exercise (walking/treadmill-based), strength training of upper and lower limbs, and educational discussions. The Polish post-COVID-19 rehabilitation program for adults includes physical efficiency training on a cycle ergometer (up to the training heart rate), walking training, breathing exercises, general fitness exercises, resistance training, station training, and relaxation [52]. As there is currently no evidence examining the effect of exercise training in children with post-COVID-19 condition, the current study intervention has been designed based on the evidence for exercise training in adults, with modifications made to suit the younger age of participants. We substituted some cardiovascular exercises for games that are designed to achieve the same intensity of exercise training but providing appeal for children aged 10–12 years. 

As this is the first study examining the effect of exercise training in children with post-COVID-19 condition, the frequency and duration of the exercise training program have also been modeled on existing evidence for exercise training in adults with chronic respiratory disease. It is unknown whether the intervention of twice weekly supervised group exercise training sessions for 8 weeks will be sufficient to induce physiological and psychological changes. In untrained healthy 12-year-old children, 8 weeks of endurance running was effective in inducing changes in VO_2_max and ventilatory threshold [53]. In adults with asthma, 12 weeks of aquatic exercises increased VO_2_max, but no improvements were demonstrated in pulmonary function [54]. However, in children, a 12-week aquatic or exergame play exercise program resulted in an increase in FVC [18]. 

The strength of the study described here is the simple design and low-cost exercise training intervention, which, aside from access to a pool or gym, requires no expensive equipment, making it feasible to implement. The interventions were designed according to four principles of effective interventions for children [55]: 1—optimize child’s engagement (through interesting equipment and creative activities); 2—provide a just-right challenge (selection of exercises and activities matching children’s skills and interests, and based on careful analysis of performance and behavior); 3—establish a therapeutic relationship (through therapist fostering a respectful, safe, and playful relationship emphasizing social group interactions); 4—provide adequate and appropriate intensity and reinforcement (rewarding system and dosage of intervention based on screening tests and age). Further, the intervention enables groups of patients to access therapy at the same time, which may have additional advantages for a child’s psychosocial wellbeing following many years of isolation due to lockdowns and periods of home schooling. 

### Limitations

At this stage of research design, we have already identified potential limitations and have tried to reduce them (e.g., access to test for antibodies against the SARS-CoV-2 coronavirus in case of a lack of positive RT-PCR test and/or positive result). However, we may face the major limitation of drop out of participants taking into account the length of the intervention (8 weeks) and frequency (2 times per week). Therefore, in an effort to avoid a high attrition rate, we have planned a reward system during the interventions and a gift (fit band) for all children after the final assessment has been completed. Moreover, the location of sport facilities (swimming pool and gym) is planned nearby a subway station to increase the convenience of public transport. We cannot also exclude the occurrence of recall bias regarding the PedsQL^TM^ 4.0 and CFSQ results; however, to minimize this, we will analyze children’s and parents’ perception of HRQOL and limit the length of the reporting period to one month of recall. Blinding will be not feasible, due to logistic and organizational issues, including limited human resources. However, recent evidence regarding the association between blinding and treatment effect estimates is still inconclusive in the field of rehabilitation [56,57]. Other limitations that will appear during the course of data collection and analysis will be identified and explained at the stage of data presentation.

## 4. Conclusions

It is important to determine the effect of exercise interventions that are feasible and effective in addressing the symptoms of post-COVID-19 condition in children. To our knowledge, this is the first study examining the effect of water-based and land-based exercise training on exercise capacity, fatigue, health-related quality of life, and pulmonary function in children aged 10–12 years with post-COVID-19 condition. We hope this study will provide guidance for long COVID-19 rehabilitation in children.

## Figures and Tables

**Table 1 ijerph-19-14476-t001:** SPIRIT table of enrolment, intervention, and assessment.

	Enrolment	Before Intervention	Randomization	Intervention	After Intervention
**TIMEPOINT**	1st and 2nd month	3rd month	3rd month	4th and 5th months (8 weeks)	6th month
**ENROLMENT**					
Eligibility pre-screening	X				
Informed consent	X				
Allocation			X		
**INTERVENTIONS**					
Water-based exercise, land-based exercise program, or control group				X	
**ASSESSMENTS**					
Demographic data	X				
Weight		X			X
Height		X			X
Diagnosis of general practitioner	X	X			
CFSQ		X			X
PedsQL^TM^ 4.0 child version		X			X
PedsQL^TM^ 4.0 parent version		X			X
Pulmonary function: VC, FVC, FEV_1_, MEF_75_, MEF_50_, MEF_25_		X			X
Exercise capacity: HR, VO_2_, VCO_2_, VE		X			X
PCERT		X		X	X

CFSQ, Cumulative Fatigue Symptoms Questionnaire; FEV_1_, forced expiratory volume in 1 s; FVC, forced vital capacity; HR, heart rate; MEF_75_, maximal expiratory flow at 75% of forced vital capacity; MEF_50_, maximal expiratory flow at 50% of forced vital capacity; MEF_25_, maximal expiratory flow at 25% of forced vital capacity; PCERT, Pictorial Children’s Effort Rating Table; PedsQL^TM^ 4.0, Pediatric Quality of Life Inventory 4.0 Generic Core Scales; SPIRIT, Standard Protocol Items: Recommendations for Interventional Trials; VC, vital capacity; VCO_2,_ carbon dioxide output; VE, minute ventilation; VO_2,_ oxygen uptake.

**Table 2 ijerph-19-14476-t002:** Water-based and land-based exercises protocol.

	Duration	Water-Based Exercise Program	Land-Based Exercise Program
**WARM-UP**	8 min	Upper- and lower-limb aerobics, including punching and kicking; jogging (stationary); breathing control; lower-limb stretches	Upper- and lower-limb aerobics, including punching and kicking; marching (stationary); breathing control; lower-limb stretches
**AEROBIC EXERCISE TRAINING**			
**1 game**	6 min		
**circuit one**	(exercise 1 min, rest 15 s) 6 min	Horizontal arm sweeps front and back (standing) Jogging in place with arms punching Holding onto the edge, lying on the front, legs working to crawl Snow angels with noodle between the legs Squat and jump	Horizontal arm sweeps front and back with dumbbells of 1 kg (standing) Jogging/marching in place with arms punching Front-back jumps with extending legs Star jumps Side to side squat with vertical arms movement
**breathing exercise**	1 min		
**1 game**	6 min		
**circuit two**	(exercise 1 min, rest 15 s) 6 min	Pushing the noodle down, arms extended (standing) Kicking in front with arms pushing the water Side to side lying with noodle under the arms Jumping over the noodle Jumping jacks	Vertical arms movement with medicine ball of 3 kg (standing) Arms punching with legs kicking Seated legs raise side to side Burpees Jumping jack
**breathing exercise**	1 min		
**1 game**	6 min		
**COOL DOWN**	5 min	Upper-limb and thoracic cage stretches; breathing control	Upper-limb and thoracic cage stretches; breathing control

**Table 3 ijerph-19-14476-t003:** Games and plays used in the water-based and land-based exercise groups.

Water-Based Exercise Program	Land-Based Exercise Program
1. Cockfight on the noodle Children matched in pairs are facing each other sitting on a noodle, children’s hands leaning against each other. The children try to throw each other off balance (the child’s task is to make the opponent fall). Each fall of the opponent is scored.	1. Cockfight Children matched in pairs are facing each other in a squat, children’s hands leaning against each other. The children, jumping in a squat try to throw each other off balance (the child’s task is to make the opponent fall). Each fall of the opponent is scored.
2. 10 ball/5 ball There are two teams. Each team makes 10/5 passes without losing the ball. The other team tries to get the ball.	2. 10 ball/5 ball There are two teams. Each team makes 10/5 passes without losing the ball. The other team tries to get the ball.
3. Fishes in the net One team of children holds hands to create the “fishing net” and move to the side. The other team (fishes) tries to jump out of the net under or over holding hands.	3. Fishes in the net Two children hold hands to form the “fishing net”; the remaining ones run. The task of the children forming the net is to catch the next fleeing children (fish). The caught “fish” joins the net, which becomes bigger and bigger.
4. Frozen tag With variations on how to be “unfrozen” (e.g., swim between legs, perform 10 jumps)	4. Frozen tag With variations on how to be “unfrozen” (e.g., crawl between legs, perform 10 squats)
5. Piggy in the middle Two children are about 10 m apart and the third child in the middle is the “piggy in the middle”. The two on the outside move around and throw the ball to each other while the one in the middle tries to catch it. If the ball is dropped, any of the three children can recover it. If the person in the middle catches or recovers the ball, the person who was the last to throw is now the “piggy in the middle”.	5. Piggy in the middle Two children are about 10 m apart and the third child in the middle is the “piggy in the middle”. The two on the outside move around and throw the ball to each other while the one in the middle tries to catch it. If the ball is dropped, any of the three children can recover it. If the person in the middle catches or recovers the ball, the person who was the last to throw is now the “piggy in the middle”.
6. Obstacle course with different variations of aerobic exercises	6. Obstacle course with different variations of aerobic exercises
7. Waiters (Cup on the board) The children are moving around in a different way (walking forward, backward, and running). Each child holds a kick board with a plastic cup filled with water placed on the board. Each participant tries to overturn the cup of the other child while not spilling their water.	7. Tail tag A “tail” (small rope) is attached to each child’s pants. Each child tries to grab the tail from the other participants.
8. Shark and sardines One child from the group is a shark and takes a place at the distance behind the sardines. At the signal, the shark chases the sardines. Caught sardines become sharks. The sardine that was last caught wins.	8. Shark and sardines One child from the group is a shark and takes a place at the distance behind the sardines. At the signal, the shark chases the sardines. Caught sardines become sharks. The sardine that was last caught wins.
9. Treasure hunters Different objects are scattered around the swimming pool. At the signal, the children are trying to collect as many objects as possible and put them in the right place. They can only collect one object at one time.	9. Treasure hunters Different objects are scattered around the gym hall. At the signal, the children are trying to collect as many objects as possible and put them in the right place. They can only collect one object at one time.
10. How many steps home All children are at one end of the pool. They ask “how many steps home?”. The leader has to invent a way of swimming (with the noodle), e.g., “3 frog movements”, “2 dolphin movements”, and “2 dives”. All have to get home using only the mentioned way of swimming.	10. How many steps home All children are in the same place. They ask “how many steps home?”. The leader has to invent a way of walking, e.g., “3 elephant steps”, “2 steps of a ballerina”, and “4 ant steps”. All have to get home using only the mentioned steps.

## Data Availability

Not applicable.

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
