# Peer review of "Water-Based and Land-Based Exercise for Children with Post-COVID-19 Condition (postCOVIDkids)—Protocol for a Randomized Controlled Trial"

_ijerph, 2022, doi:10.3390/ijerph192114476_

Round 1

Reviewer 1 Report

Thank you to the authors for the opportunity to review this manuscript – this article provides a detailed description of a prospective study focused on interventions to increase physical activity and exercise among children with post-COVID symptoms, specifically seeking to compare land-based and water-based forms of physical activity and their impact in addressing post-COVID symptoms and general health and wellness. In my view, this article is well-organized and presents a thoroughly designed prospective study; the introduction and review of literature do very well to clearly state the primary issue and object of the study, and the prospective study methodology is comprehensively addressed. My comments below reflect several minor issues that could be addressed to improve this manuscript for publication:

1)      One issue for the authors to consider, and potentially address in the Limitations section, would be possible recall error or recall bias in regard to the use of surveys and questionnaires – given that this data would be collected only at the start and finish of the physical activity program, there may be possible recall error from a period of 8 weeks. However, this issue is in part addressed through the resolving of parent and student data as described in the manuscript.

2)      Pg 2 line 59-60, revise (“also an organisational problem”), and could include an additional statement on why this issue is important in the context of this study

3)      Pg 4 line 147-148, revise (“will be measured”)

4)      Pg 9 line 193-194, revise (two periods after “researcher”)

Author Response

 Reply to Reviewers

Water and land-based exercise for children with post COVID-19 condition (postCOVIDkids) – protocol for a randomized controlled trial

(Manuscript ID ijerph- 1982223)

The authors thank the Reviewers for the examination of our paper. We trust our explanations and additions to the paper sufficiently address the reviewer concerns and suggestions and the revised manuscript aligns well with The International Journal of Environmental Research and Public Health publication goals.

Reviewer 1

Thank you for the careful examination of our paper. We trust our explanations and additions to the paper sufficiently address your concerns and suggestions.

Thank you to the authors for the opportunity to review this manuscript – this article provides a detailed description of a prospective study focused on interventions to increase physical activity and exercise among children with post-COVID symptoms, specifically seeking to compare land-based and water-based forms of physical activity and their impact in addressing post-COVID symptoms and general health and wellness. In my view, this article is well-organized and presents a thoroughly designed prospective study; the introduction and review of literature do very well to clearly state the primary issue and object of the study, and the prospective study methodology is comprehensively addressed. My comments below reflect several minor issues that could be addressed to improve this manuscript for publication:

1)      One issue for the authors to consider, and potentially address in the Limitations section, would be possible recall error or recall bias in regard to the use of surveys and questionnaires – given that this data would be collected only at the start and finish of the physical activity program, there may be possible recall error from a period of 8 weeks. However, this issue is in part addressed through the resolving of parent and student data as described in the manuscript.

We have addressed the recall bias possibility in the limitation section.

2)      Pg 2 line 59-60, revise (“also an organisational problem”), and could include an additional statement on why this issue is important in the context of this study

It was explained.

3)      Pg 4 line 147-148, revise (“will be measured”)

 It was corrected.

4)      Pg 9 line 193-194, revise (two periods after “researcher”)

It was corrected.

Reviewer 2 Report

Comments

Introduction lacks information about land-based exercise programs in children and its impact on exercise capacity.

Please add information on how do you plan to monitor attendance rate (2.5 Procedure). What attendance rate in exercise program is planned to be considered for analysis (2.7 Missing data)?

Some additional editing comments:

The tens of manuscript parts regarding planned research should be future (lines 107 and 150).

Table 1: There is lack of shortcut FVC description under the table. I would also recommend to add description of SPIRIT shortcut below the table to make sure it is self-explanatory and the reader will be able to understand the contents of the table without referring to the text. The format of shortcuts regarding subscripts should be unified through the manuscript – e.g. MEF50 or MEF50

The standard units of measurement defined by the International System of Units (SI) recommends using symbol s for seconds – please consider to change it through the manuscript.

Line 165: Please reject additional full stop and comma at the end of sentence.

Line 195: Please indicate what formula is planned to calculate the body surface area .

Line 208: psychologicaloverload should be written separately.

Line 240: Why absolute values of Vital Capacity (VC) is not mentioned?

Line 275: the dash sign seems to be unnecessary

Line 290: Maybe ‘program’ will be better term here than ‘group’.

Line 360: Please reject additional full stop at the end of sentence.

Line 407: There is a space missing

Line 411: delete “been” from the sentence

Author Response

Reviewer 2

Thank you for the careful examination of our paper. We trust our explanations and additions to the paper sufficiently address your concerns and suggestions.

Comments

Introduction lacks information about land-based exercise programs in children and its impact on exercise capacity.

The information was added.

Please add information on how do you plan to monitor attendance rate (2.5 Procedure). What attendance rate in exercise program is planned to be considered for analysis (2.7 Missing data)?

The information was added in the procedure section. We do not plan to assume particular attendance rate, rather aim to focus on what attendance rate is needed to observe incremental of outcomes.

Some additional editing comments:

The tens of manuscript parts regarding planned research should be future (lines 107 and 150).

It was corrected.

Table 1

There is lack of shortcut FVC description under the table. I would also recommend to add description of SPIRIT shortcut below the table to make sure it is self-explanatory and the reader will be able to understand the contents of the table without referring to the text. The format of shortcuts regarding subscripts should be unified through the manuscript – e.g. MEF50 or MEF50

It was corrected.

The standard units of measurement defined by the International System of Units (SI) recommends using symbol s for seconds – please consider to change it through the manuscript.

It was corrected.

Line 165: Please reject additional full stop and comma at the end of sentence.

It was corrected.

Line 195: Please indicate what formula is planned to calculate the body surface area.

It was added.

Line 208: psychologicaloverload should be written separately.

It was corrected.

Line 240: Why absolute values of Vital Capacity (VC) is not mentioned?

It was added.

Line 275: the dash sign seems to be unnecessary

It was corrected.

Line 290: Maybe ‘program’ will be better term here than ‘group’.

It was corrected.

Line 360: Please reject additional full stop at the end of sentence.

It was corrected.

Line 407: There is a space missing

It was corrected.

Line 411: delete “been” from the sentence.

It was corrected.